# Resection of a Solitary Right Ventricular Metastasis in Oligorecurrent Hepatocellular Carcinoma

**DOI:** 10.3390/jcm12247530

**Published:** 2023-12-06

**Authors:** Defne Gunes Ergi, Kyle W. Klarich, Joseph A. Dearani, Juan A. Crestanello, Joseph J. Maleszewski, Jonathan M. Morris, Phillip M. Young, Cameron M. Callaghan, Henry C. Pitot, Arman Arghami

**Affiliations:** 1Department of Cardiovascular Surgery, Mayo Clinic, Rochester, MN 55901, USA; gunes_ergi@hotmail.com (D.G.E.); jdearani@mayo.edu (J.A.D.); crestanello.juan@mayo.edu (J.A.C.); 2Department of Cardiology, Mayo Clinic, Rochester, MN 55901, USA; klarich.kyle@mayo.edu; 3Department of Laboratory Medicine and Pathology, Mayo Clinic, Rochester, MN 55901, USA; maleszewski.joseph@mayo.edu; 4Department of Radiology, Mayo Clinic, Rochester, MN 55901, USA; morris.jonathan@mayo.edu (J.M.M.); young.phillip@mayo.edu (P.M.Y.); 5Department of Radiation Oncology, Mayo Clinic, Rochester, MN 55901, USA; callaghan.cameron@mayo.edu; 6Department of Medical Oncology, Mayo Clinic, Rochester, MN 55901, USA; pitot.henry@mayo.edu

**Keywords:** cancer, metastasis, hepatocellular cancer, cardiac surgery

## Abstract

Hepatocellular carcinoma (HCC), constituting the predominant manifestation of liver cancer, stands as a formidable medical challenge. The prognosis subsequent to surgical intervention, particularly for individuals presenting with a solitary tumor, relies heavily on the degree of invasiveness. The decision-making process surrounding therapeutic modalities in such cases assumes paramount importance. This case report illuminates a rather unusual clinical scenario. Here, we encounter a patient who, following a disease-free interval, manifested an atypical presentation of HCC, specifically, a solitary cardiac metastasis. The temporal interval of remission adds an additional layer of complexity to the case. Through a multidisciplinary planning process, the decision was made for surgical removal of the metastatic tumor.

## 1. Introduction

Hepatocellular carcinoma (HCC) predominates as the primary hepatic malignancy, characterized by a 5-year survival rate of 18%, positioning it as the second most fatal neoplasm, surpassed solely by pancreatic cancer [1]. The incidence of cardiac metastasis in HCC is documented within the range of 5% to 10% [2]. Therapeutic modalities in this context encompass chemoembolization and debulking surgery [3]. The gravitas of metastatic manifestations necessitates a careful selection of therapeutic interventions, contingent upon patient preferences and prognostic considerations. 

This case report delineates the clinical trajectory of a patient initially afflicted by localized hepatocellular carcinoma, who achieved a commendable remission duration extending nearly 5 years. Subsequently, the emergence of a metastatic lesion within the right ventricle (RV) presented a clinical challenge. Following multidisciplinary assessments, the consensus leaned towards the surgical debulking operation. This choice was driven by the objectives of alleviating symptomatic burdens and enhancing the patient’s quality of life. 

### 1.1. Case Report

A 74-year-old woman, who had five years prior undergone a partial hepatectomy for hepatocellular carcinoma, was referred to our institution for evaluation of an 8 × 5 × 6 cm^3^ mass located within the RV. She did not have an elevated alpha-fetoprotein in her history which was negative all the time even prior to hepatic resection and was not treated post-surgical intervention with chemotherapy. In 2018, preoperative magnetic resonance imaging (MRI), demonstrated hepatic mass measuring approximately 13.0 × 10.0 cm^2^ in size involving posterior segment of the right liver lobe (Figure 1). Her initial surgery for localized HCC had resulted in clear margins; however, surgical report did mention some evidence of vascular invasion. She has been surveilled every 3 months for 2 years, and every 6 months since then with no recurrence. On her recent surveillance in this year, a computed tomography (CT) scan revealed the presence of an RV mass, accompanied by a thrombus within the left brachiocephalic vein after which she was started on apixaban. During this period, she had a significant decline in her exertional tolerance to the point where she had to move around by wheelchair. She developed significant orthopnea and dyspnea on exertion which eventually required her to visit the emergency department. 

Upon arrival at our institution, she was experiencing tightness accompanied by shortness of breath. In addition to apixaban, she was also on medication for bipolar disorder and a calcium channel blocker for hypertension. Notably, she was not receiving chemotherapy. Cardiac MRI revealed a sizable RV free wall mass (7.9 × 5.3 × 3.0 cm^3^), protruding into the right ventricular outflow tract (RVOT) causing severe obstruction. An endomyocardial biopsy was conducted, and the diagnosis revealed metastatic HCC. 

Due to severity of her symptoms and the fact that most of the symptoms were caused by RVOT blockage, it was believed she would not respond well to slower treatments like chemotherapy, radiotherapy, or chemoembolization. As a result, she was offered debulking surgery. For surgical planning, she underwent a dedicated CT angiogram with 3D reconstruction and 3D model printing. Additionally, she underwent a coronary angiogram, which revealed the presence of multiple large collaterals supplying the tumor. Given the existence of multiple feeding vessels, we decided against performing embolization. 

### 1.2. Three Dimensional Printing Process for the Tumor

As shown in Figure 2 and Figure 3, the 3D printing process for the tumor had several sequential steps. After the imaging process, the 0.2 mm slices were imported into Mimics 24.0 (Materialise, Leuven, Belgium). The segmentation of different anatomical structures was executed through a combination of thresholding and manual segmentation techniques. The segmented structures encompassed the aorta and coronary arteries, blood pools of both the left and right atrium, myocardium of the left and right ventricles, aorta, pulmonary veins, pulmonary arteries, and the tumor itself. The manual segmentation process was a collaborative effort involving an experienced radiologist and a cardiac radiologist working together to ensure accuracy and precision in identifying and delineating these structures.

### 1.3. Surgical Procedure

The preoperative transesophageal echocardiogram demonstrated severely reduced right ventricular systolic function with thickened tricuspid valve (TV) and mild–moderate TV regurgitation. Left ventricular ejection fraction (LVEF) was 60%. The patient was placed on cardiopulmonary bypass, and the body temperature was lowered to 34 °C. The heart was arrested using intentionally higher-than-normal dose of del Nido cardioplegia solution due to the tumor’s significant blood flow through the right coronary artery (RCA). Upon inspecting the TV, a small flail portion of the anterior leaflet was also discovered. Retractors were used to retract the tricuspid leaflet, exposing the tumor. The head of the tumor that extended into the RVOT exhibited limited adhesions to the RVOT, while the rest of the tumor was firmly adhered to the RV free wall. From the external side, the greenish tumor was visible just beneath a thin epicardial layer covering a moderate area (Figure 4A). This raised concerns that debulking the tumor might result in a major defect requiring patching, potentially causing significant RV dysfunction. Therefore, we limited the tumor debulking over the areas where the tumor had crossed the free wall. The tumor possessed a partial capsule but exhibited multiple areas where it extended beyond the capsule, forming greenish protrusions (Figure 4B,C). The RVOT position of the tumor was then carefully resected in one piece with its capsule. After entering the capsule, the remaining tumor in the RV was found to be fragile-looking green tumor cells that was easily suctioned using a throw away sucker (Figure 4D). Throughout the procedure, care was taken to keep pump suckers away from tumor cells, and instrument sets were changed at the end after thorough irrigation to minimize cross-contamination. Using a combination of scissors and suction, about 80% of the tumor was debulked, leaving around 20% near the apex and some in proximity to the RCA. The extracted segments of the tumor were forwarded to the surgical pathology laboratory for in-depth analysis (Figure 4E,F and Figure 5). 

Given that a complete R0 resection was not feasible without risking significant heart failure and RV dysfunction, a decision was made not to create a large defect in the RV that could lead to unrecoverable complications. As a result, the procedure was concluded at this point. The area was extensively irrigated with saline, and potential cell remnants were removed. The TV was repaired with a 26 mm contour 3D partial rigid band. The flail segment of the anterior leaflet was then plicated using two figure-of-eight Prolene sutures. After confirming the absence of a patent foramen ovale, the cross clamp was removed, the RA was de-aired, and it was closed.

A prophylactic anterior pericardiotomy was performed mainly due to the patient’s need for radiation therapy and the size of the tumor. The patient was successfully weaned off cardiopulmonary bypass. Postoperative TEE confirmed good left ventricular (LV) function, residual moderate RV dysfunction, no RVOT obstruction, and trivial TV regurgitation with a gradient of 2 mm Hg. The LVEF was 75%. 

The patient had a satisfactory recovery during the postoperative period and was discharged four days following the surgery. She was referred to cardiac rehabilitation and prescribed furosemide, bisoprolol, and warfarin. During the most recent follow-up at one month, the patient remains asymptomatic. 

## 2. Discussion

In this complex clinical scenario, we detailed our surgical approach to managing a cancer patient and highlighted the crucial role of personalized decision-making. This case presented a unique challenge that required the involvement of various specialists, including those from cardiology, radiology, oncology, and cardiovascular surgery. After a careful assessment, it was determined that surgical removal of the tumor mass would offer the greatest potential benefit to the patient. However, it was essential to ensure that the patient had an understanding of the surgical procedure, which aimed to reduce the tumor burden rather than provide a curative outcome. The patient’s informed consent and willingness to proceed with the surgery were important, underscoring the significance of effective communication and shared decision-making.

Determining the mechanism of spread to the heart can be challenging in these cases. Most cardiac metastases often result from the extension of intrahepatic HCC that travels to the inferior vena cava (IVC), and isolated cardiac metastases such as in the present case, are uncommon [4]. Previously reported cases of cardiac metastasis of HCC showed different localizations. Kawakami and colleagues reviewed 17 cases in the literature with isolated cardiac metastasis of HCC. The majority of the patients were symptomatic (88.2%), and 76.5% had dyspnea. The localizations of the metastatic tumors included mostly RV (n = 10, 58.8%), followed by RA (n = 5, 29.4%) and LV (n = 2, 11.8%) [5]. 

In the present case, patient’s previous cancer status in 2018 was ‘‘pT2Nx’’, indicating a 10 cm primary mass with negative margins and moderate differentiation. It was updated with recurrence with metastasis to the RV and the current status involved a maximal safe subtotal resection with residual disease in the basal 2/3 of the RV free wall, approximately 1.3 cm in thickness, which was deemed unsafe to remove. Focus should be placed on avoiding RV free wall disruption when the procedure is not an R0 resection. If necessary, one can use pericardial patch to repair the RV defect with interrupted pledgeted large sutures to repair the RV free wall in a tension free manner. Our patient was scheduled for radiation therapy and possible systemic treatment after the healing period. With that in mind, we elected to perform a limited prophylactic pericardiectomy to prevent future pericardial disease from radiation. This decision was based on recent studies that emphasize the effects of radiation even at low doses on cardiac events [6,7].

## 3. Conclusions

In advanced-stage cancer patients, palliative care is often preferred, to improve quality of life. In the present case, debulking surgery offered the most effective palliation by providing immediate symptom relief and significantly increasing the chances of long-term local disease control.

## Figures and Tables

**Figure 1 jcm-12-07530-f001:**
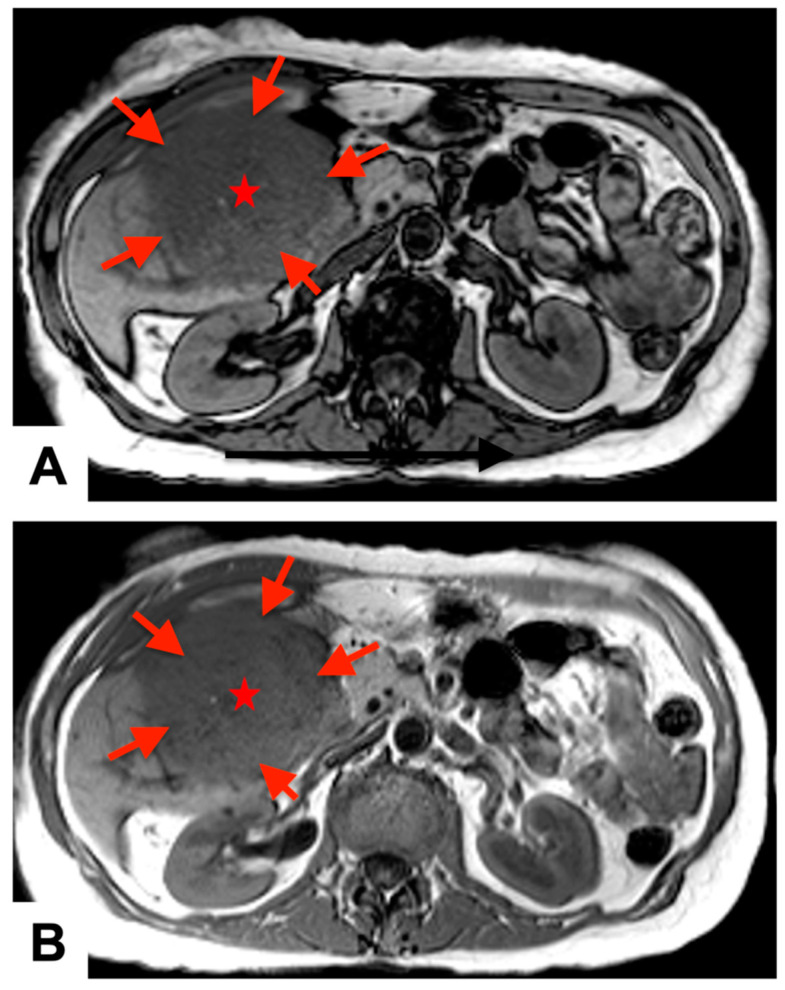
Preoperative magnetic resonance imaging: T1 (**A**) and T2 (**B**) axial phase views demonstrate a large, heterogeneously enlarging, exophytic hepatic mass (red arrows and stars) in the posterior segment of the right lobe of the liver, measuring approximately 13.0 × 10.0 cm^2^ in size.

**Figure 2 jcm-12-07530-f002:**
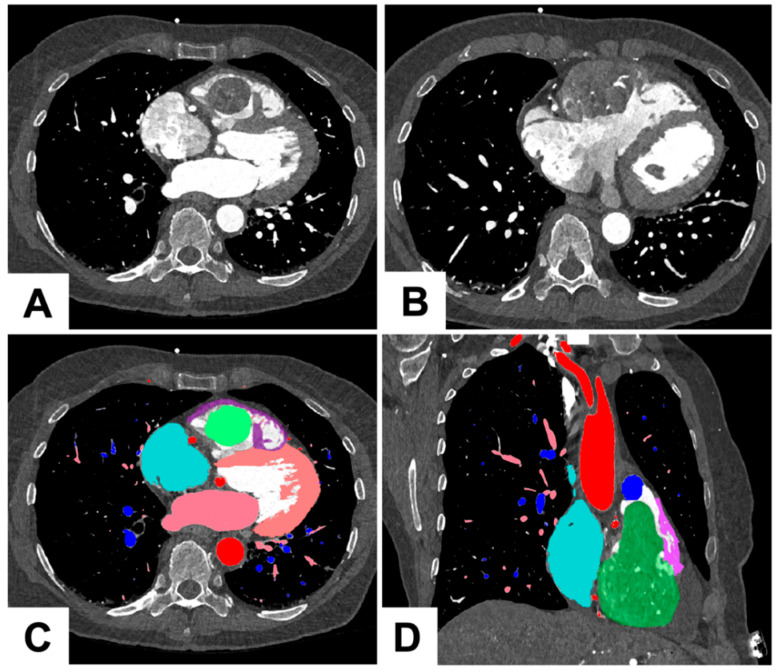
Illustration of preoperative 3D tumor excision planning. (**A**) Axial slice: Image illustrating the tumor’s pedunculated section as it approaches the RVOT; (**B**) Axial slice: Slice demonstrating tumor extending through the myocardium of the RV; (**C**) Color-coded visualization: The tumor mass is indicated in green, the right ventricle in purple, the LV in peach, the RA in cyan, the left atrium and pulmonary veins in pink, the pulmonary arteries in blue and the aorta in red; (**D**) Coronal section: The view along the tricuspid plane, displaying the tumor’s pedunculation extending through the RVOT.

**Figure 3 jcm-12-07530-f003:**
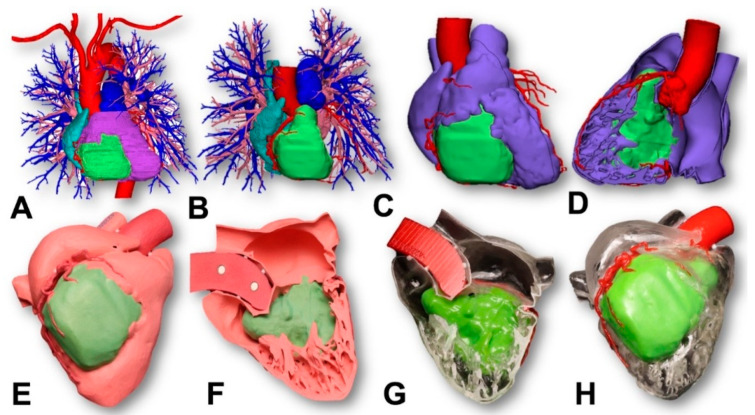
Step-by-step demonstration of the 3D printing process. (**A**) Initial 3D file export: The first step involved exporting the 3D file to the medical computer-aided diagnosis (CAD) software 3-Matic 16.0 (Materialise, Leuven, Belgium); (**B**) CAD file refinement: Subsequently, the CAD file underwent a smoothing process, eliminating 3D file errors, and optimizing it for the printing process; (**C**,**D**) Removal of unwanted anatomy: Unnecessary anatomical components were removed from the model before printing. The RV, RA, and pulmonary artery were merged in the CAD software for printing. Additionally, as per the surgeon’s request, the model was sectioned along the RVOT. This was followed by further polygon count reduction through additional model smoothing; (**E**,**F**) Dual printing approach: The first model was produced using a color nylon HP printer, segmented with a magnet casing to enable separation of the RV components; (**G**,**H**) Polyjet printing on Stratasys J5: The second model was crafted using a Polyjet Stratasys J5 printer, utilizing both clear and color materials to visualize the tumor’s extent.

**Figure 4 jcm-12-07530-f004:**
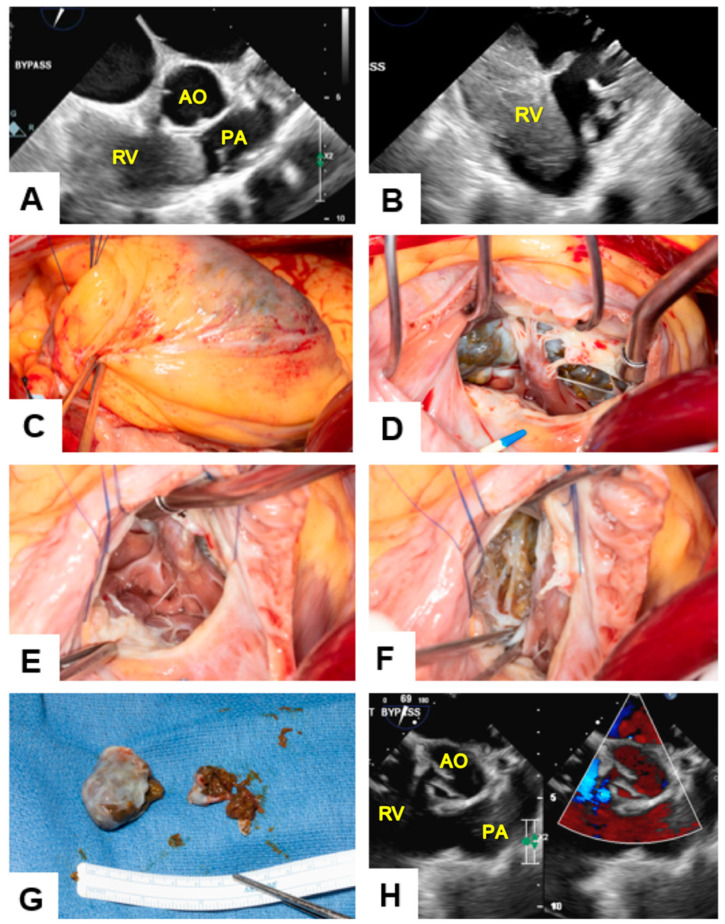
Illustration of intraoperative images of mass excision surgery. (**A**,**B**) Preoperative transesophageal echocardiogram: Image displaying the tumor mass in the RV obstructing the pulmonary artery outflow tract, as observed through transesophageal echocardiography (Image coding: RV = Right ventricle, AO = Aorta, PA = Pulmonary artery); (**C**) External view of tumor mass: This image shows an external perspective of the tumor mass, showing a greenish area on the surface, as observed from the surface of the RV (Image coding: RV = Right ventricle); (**D**,**E**) Tumor presentation prior to excision: (**D**) Image showing the tumor before its excision, revealing the presence of brown-green colored material on its surface; (**E**) Image demonstrating a widely opened RVOT, revealing that tumor cells were not adherent to the RVOT, enabling excision from the TV without accessing the RVOT; (**F**) Residual right ventricle wall: Image shows the aftermath of tumor excision. It displays the remaining wall of the RV with residual tumor cells attached to the RV free wall; (**G**) Surgical explant materials: These images show the materials extracted during surgery, for pathological examination. Due to the use of a throw away sucker, only a portion of the resected tumor cells is visible; (**H**) Postoperative transesophageal echocardiogram: Image from a postoperative view, displaying a relieved pulmonary outflow tract on the left-hand side and uninterrupted flow passing through without any obstruction, as observed in Doppler mode on the right-hand side (Image coding: RV = Right ventricle, AO = Aorta, PA = Pulmonary artery).

**Figure 5 jcm-12-07530-f005:**
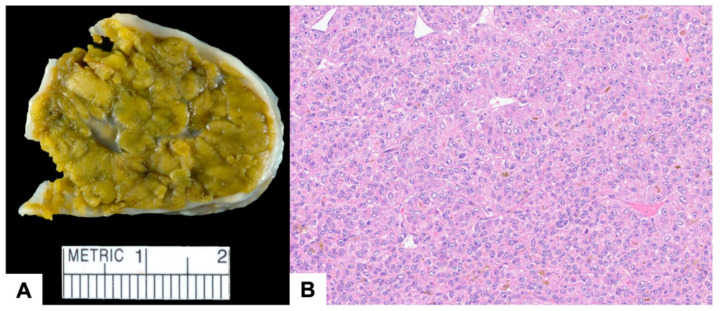
(**A**) Gross pathological examination revealed a well-defined mass with a brown-green, somewhat lobulated, cut surface. (**B**) Histologically, the mass consisted of sheets of malignant epithelioid hepatocytes, with areas showing the presence of brown bile pigment.

## Data Availability

Data are contained within the article.

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
