# Peer review of "Resection of a Solitary Right Ventricular Metastasis in Oligorecurrent Hepatocellular Carcinoma"

_jcm, 2023, doi:10.3390/jcm12247530_

Round 1

Reviewer 1 Report

Comments and Suggestions for Authors

The authors reported an interesting case of hepatocellular carcinoma with isolated metachronous cardiac metastasis 5 years after initial diagnosis.

The case has been well-written and discussed with good documentation.

Minor comments:

1. In Figure 1, it is better to mark the liver mass with arrows around it.

2. In Figure 3, change  "adided" to "aided" 

3. In the case report section, clarify how you excluded synchronous metastases.

Author Response

Reviewer 1

The authors reported an interesting case of hepatocellular carcinoma with isolated metachronous cardiac metastasis 5 years after initial diagnosis. The case has been well-written and discussed with good documentation. Minor comments:

  1. In Figure 1, it is better to mark the liver mass with arrows around it.

Response:

Your thoughtful comments on the manuscript and feedback on Figure 1 are greatly appreciated. We have made revisions to the figure based on your input.

  1. In Figure 3, change "adided" to "aided"

Response

Thank you for pointing this, we revised Line 91 as ‘’: The first step involved exporting the 3D file to the medical computer-aided diagnosis (CAD) software 3-Matic 16.0 (Materialise, Leuven, Belgium);’’

  1. In the case report section, clarify how you excluded synchronous metastases

Response

CT scans were used to check for synchronous metastases as part of the preoperative preparations for debulking surgery.

Reviewer 2 Report

Comments and Suggestions for Authors

This is a very compelling case summary with an excellent narrative and robust complementary imaging.  The manuscript documents an individualized oncologic approach to a solitary cardiac metastasis of hepatocellular origin in a 74 y/o female patient.

The complex, multidisciplinary decision-making for oncologic treatment, preoperative planning, and incorporation of novel technology for a comprehensive strategy will be of interest to a broad range of clinicians. 

I think this is an excellent submission. If it were targeting a primarily surgical journal, some further explanations of technique, approach, pitfalls & pearls, and bailout strategy if the RV free wall were compromised might be of interest.

Author Response

Reviewer

This is a very compelling case summary with an excellent narrative and robust complementary imaging.  The manuscript documents an individualized oncologic approach to a solitary cardiac metastasis of hepatocellular origin in a 74 y/o female patient. The complex, multidisciplinary decision-making for oncologic treatment, preoperative planning, and incorporation of novel technology for a comprehensive strategy will be of interest to a broad range of clinicians. I think this is an excellent submission.

Response

Your thoughtful comments on the manuscript are greatly appreciated.

If it were targeting a primarily surgical journal, some further explanations of technique, approach, pitfalls & pearls, and bailout strategy if the RV free wall were compromised might be of interest.

Response

Focus should be placed on avoiding RV free wall disruption when the procedure is not an R0 resection. If necessary, one can use pericardial patch to repair the RV defect with interrupted pledgetted large sutures to repair the RV free wall in a tension free manner. 

Reviewer 3 Report

Comments and Suggestions for Authors

I had the pleasure of reviewing this interesting case report by Ergi et al., in which they described the successful subtotal resection of an HCC metastasis infiltrating the right ventricle. The study is very interesting, well-executed, and well-written. They highlighted the importance of personalized treatment through the involvement of several specialists, involved in the entire course, from diagnosis to the prediction and prevention of possible complications due to the treatment.

Author Response

Reviewer 

I had the pleasure of reviewing this interesting case report by Ergi et al., in which they described the successful subtotal resection of an HCC metastasis infiltrating the right ventricle. The study is very interesting, well-executed, and well-written. They highlighted the importance of personalized treatment through the involvement of several specialists, involved in the entire course, from diagnosis to the prediction and prevention of possible complications due to the treatment.

Response

Your thoughtful comments on the manuscript are greatly appreciated. Thank you.
